# Diabetes-Related Mechanisms of Action Involved in the Therapeutic Effect of *Croton* Species: A Systematic Review

**DOI:** 10.3390/plants12102014

**Published:** 2023-05-18

**Authors:** Fernanda Artemisa Espinoza-Hernández, Angelina Daniela Moreno-Vargas, Adolfo Andrade-Cetto

**Affiliations:** 1Laboratorio de Etnofarmacología, Facultad de Ciencias, Universidad Nacional Autónoma de México, Ciudad Universitaria, Coyoacán, CDMX C.P. 04510, Mexico; 2Posgrado en Ciencias Biológicas, Unidad de Posgrado, Edificio D, 1° Piso, Circuito de Posgrados, Ciudad Universitaria, Coyoacán, CDMX C.P. 04510, Mexico

**Keywords:** traditional medicine, medicinal plants, *Croton* genus, type 2 diabetes, action mechanisms

## Abstract

Over the years, ethnopharmacological and phytochemical investigations have been conducted to understand the potential effects of the *Croton* genus on several diseases. It has been revealed that these terpenoid-rich species traditionally used to treat gastrointestinal diseases, heal wounds, and relieve pain have a wide range of therapeutic effects; however, those used to treat diabetes, as well as their action mechanisms, have not been reviewed so far. Therefore, the main objective of this review was to compile all *Croton* species that have shown pharmacological effects against diabetes and describe their action mechanisms. Through a search of the literature, 17 species with hypoglycemic, antihyperglycemic, antilipidemic, antihypertensive, antioxidant, and anti-inflammatory effects were found. Among the mechanisms by which they exerted these effects were the inhibition of α-glucosidases, the promotion of insulin secretion, and the increase in glucose uptake. Interestingly, it was found that some of them may have antihyperglycemic properties, although there were no ethnopharmacological reports that support their traditional use. Moreover, others only presented studies on their hypoglycemic effect in fasting, so further works are encouraged to describe the mechanisms involved in lowering fasting blood glucose levels, such as hepatic glucose production, especially for *C. cajucara*, *C. cuneatus*, *C. gratissimus* var. *gratissimus*, *C. guatemalensis*, and *C. membranaceus*. It is expected that this review contributes to the plant science knowledge of the genus, and it can be used in future references on the identification and development of new molecules/phytomedicines that help in the treatment of diabetes.

## 1. Introduction

Plants remain a crucial resource for helping humanity treat various types of diseases using two main approaches. The first is ethnobotany, the study of how plants are associated with their observed traditional effects, while the other is bioprospecting, by which new pharmacological entities are identified and modified using plants and their natural products as starting points. Newman et al. reported that, between 1981 and 2019, there were at least 35 chemical entities identified from natural sources or modified/synthetized based on natural products that have potential use in the treatment of diabetes [1]. Specifically, plants have proven to be interesting sources of natural products with effective mechanisms of action for the management of diseases such as diabetes [2]. In fact, the use of 1300 plant species for the treatment of this disease worldwide have been reported [3]. Therefore, plant science can play an important role in the discovery of new molecules that can contribute to the management of complex diseases.

Diabetes mellitus (DM) is defined as a group of metabolic diseases characterized by chronic hyperglycemia resulting from defects in insulin secretion or action. According to the International Diabetes Federation, the incidence of diabetes is increasing worldwide. In 2021, there were 537 million people with this condition, and there will be an estimated 783 million people diagnosed with diabetes in 2045 [4]. Therefore, DM has become one of the most prevalent noncommunicable diseases in recent decades.

Type 2 diabetes (T2D) is the most relevant type of diabetes since it represents approximately 90% of all cases [5]. The main characteristic of T2D is insulin resistance (IR), a pathological state in which there is an altered response to insulin by insulin-sensitive tissues, mainly the skeletal muscle, adipose tissue, and liver. Being overweight, or obesity, trigger low-grade inflammatory environments that cause a decreased insulin response in target organs. These conditions favor proinflammatory states that consequently lead to oxidative stress (OxS). Both proinflammatory cytokines and reactive oxygen species (ROS) induce an altered response in insulin signaling that progresses to IR [6].

In developing countries around the world, people with diabetes usually resort to the use of plant remedies as part of their traditional medicine (TM) instead of or in addition to conventional drugs for treating the disease and its associated complications [7]. According to the World Health Organization (WHO), 88% of all WHO member states use traditional and complementary medicine for the maintenance of health and the treatment of illnesses [8].

One of the genera that presents recognized species for their toxic/medicinal properties, their high chemical diversity, and their wide usage as herbal remedies is *Croton*. This genus, which belongs to the Euphorbiaceae family, includes approximately 1300 species of herbs, shrubs, and trees distributed worldwide in tropical and subtropical areas. Many of them are popularly used to treat gastrointestinal ailments, to relieve pain, for wound healing, and as malaria and dysentery remedies in the TM of Africa, Asia, and the Americas. In addition, they have been reported to be used in the treatment of complex diseases such as cancer and diabetes [9,10].

Furthermore, a great variety of secondary metabolites have been identified from *Croton* species, such as diterpenoids, triterpenoids, steroids, lignoids, alkaloids, flavonoids, and proanthocyanidins. However, diterpenoids are the most common compounds found in the extracts of these species. Diterpenoids show a wide range of hypoglycemic and antihyperglycemic properties, such as enzyme inhibition of α-glucosidases, α-amylases, protein tyrosine phosphatase 1B, and dipeptidyl peptidase IV; interaction with peroxisome proliferator-activated receptors γ; and the promotion of insulin secretion and glucose uptake via glucotransporter 4 (GLUT4) translocation [11]. One of these, *trans*-dehydrocrotonin (t-DCTN), a clerodane diterpene isolated from *C. cajucara* Benth., stands out due to its well-documented hypoglycemic, hypolipidemic, and anticancer effects [10].

Despite the great importance of the *Croton* species for assorted traditional uses and the attempts to revise the phytochemistry and pharmacology of the genus [10,12,13,14,15], the specific action mechanisms on diabetes have not been reviewed to date. The works that have been published summarize all the usages and pharmacological studies by species, so a review that includes all the species with reports on diabetes has not been carried out. According to these works, only the in vivo hypoglycemic and antihyperglycemic effects of some *Croton* species have been assessed without addressing studies on their mechanisms. Therefore, the main objective of this review was to compile all *Croton* species and their identified compounds that exhibited any pharmacological effect related to diabetes, describe the mechanisms of action involved, and provide a state-of-the-art review on the topic. In addition, this review aimed to contribute to the plant science knowledge of this genus.

## 2. Results

A total of 17 species with pharmacological reports related to diabetes and associated metabolic diseases were identified. Table 1 provides a summary of the diabetes-related mechanisms of action of all the species found, while Figure 1 compiles the chemical structures of terpenoids identified in these species with reported diabetes-related pharmacological activities.

### 2.1. C. bonplandianus Baill

Common names: “escoba negra”, “ban tulsi”. Species native to South American countries (from Bolivia to Uruguay) that was introduced into Bangladesh, India, Pakistan, and Thailand. A decoction of a handful of leaves in two liters of water is prepared and consumed as “agua de uso” (a tea that is drunk during the day) for liver ailments by the Criollos of northwestern Argentine Chaco province [16].

Even though there are no ethnopharmacological reports directly related to diabetes for this species, some authors assessed its potential antihyperglycemic effect by inhibiting the activity of carbohydrate-degrading enzymes. For instance, Qaisar et al. tested the potential inhibitory effect on α-glucosidases of the methanolic and dichloromethane extracts of the whole plant. The results show that only the dichloromethane extract inhibits the activity of these enzymes, displaying an IC_50_ of 14.93 μg/mL, which is more potent than that observed by acarbose control (IC_50_ = 38.25 μg/mL) [17]. Years later, through a bioactivity-guided isolation approach, the same group identified the compounds responsible for the α-glucosidase inhibitory effect (2 H-1-benzopyran-2-one, betulin (Figure 1a), and 3,5-dimethoxy 4-hydroxy cinnamic acid), which exhibited IC_50_ values ranging from 23 to 26.7 μg/mL [18]. It is important to note that the complete extract presented a more potent inhibitory effect than the compounds alone.

Karuppiah Vijayamuthuramalingam et al. reaffirmed the potential antihyperglycemic effect of this species by assessing the inhibitory effect of the methanolic leaf extract and its fractions on α-glucosidase and α-amylase enzyme activities. The results show that the chloroform fraction has the lowest IC_50_ values on both enzymes, which are comparable to those observed in the control. In addition, chloroform fraction exhibits a higher content of polyphenols, flavonoids, and tannins than the other fractions. Specifically, the presence of Z-5-nonadecene, cyclotetracosane, N-nonadecenol-1, cycloeicosane 3-eicosene, Z-8—hexadecene 6-5heptadecenal, and phenol,2,4-bis(1,1-dimethyl) was reported [19].

### 2.2. C. cajucara Benth

Common name: “sacaca”. Species native to the Amazonian region of Brazil. A tea is prepared by boiling 25 g of the stem bark in one liter of water and then a cup is drunk twice a day. This species is used for the treatment of diabetes, liver and kidney disorders, and to lower blood cholesterol. In addition, the use of the leaves is recommended for overweight people to lose weight [49].

Various studies showed the effectiveness of this species and its main compound, the t-DCTN (Figure 1b), a 19-nor-clerodane diterpene isolated from the bark, on glucose and lipid metabolism as well as cardiovascular and OxS risk factors. For instance, the hypoglycemic and antihyperglycemic effects of t-DCTN were demonstrated in alloxan-induced hyperglycemic rats by significantly reducing fasting blood glucose levels by 53% and significantly diminishing the hyperglycemic peak after a glucose load at a dose of 50 mg/kg [20]. In addition, microspheres of t-DCTN have been made in order to offer a potential delivery system and enhance the pharmacokinetic profile of this compound [50].

On the other hand, the antihypertriglyceridemic effect of t-DCTN was proved in ethanol-induced hypertriglyceridemic rats by causing a significant 37% decrease in this parameter at 50 mg/kg [21]. In the same way, the antihypertensive effect of this compound was tested in normotensive rats by reducing the arterial pressure and heart rate after intravenous bolus injections of t-DCTN at 10 and 15 mg/kg [22]. Finally, the aqueous extract of the *C. cajucara* bark has also been shown to reduce hepatic OxS markers in streptozotocin-induced hyperglycemic rats, which could be useful in the treatment of T2D in the long-term [23,24].

### 2.3. C. cuneatus Klotzsch

Common name: “arapurima”. Species native to South America that is used in the Venezuelan TM as an anti-inflammatory remedy, which can be mixed with cane liquor [31]. According to Torrico et al., it is also used for treating diabetes by indigenous people in Venezuela [25]. In this work, the aqueous extract of the stem bark showed that it significantly decreased blood glucose levels from the first two hours of its intraperitoneal administration at all doses proved (6.4, 13, and 52 mg/kg) without exhibiting a dose-dependent effect.

### 2.4. C. ferrugineus Kunth

Common name: “mosquera”. Species native to Brazil, Colombia, Ecuador, and Peru. As well as *C. cuneatus*, this species is used as an anti-inflammatory remedy in the Ecuadorian TM [51]; however, there is no ethnopharmacological report that directly relates it to diabetes. It has been reported that the essential oils of *Croton* species have different biological activities; therefore, Valarezo et al. tested the antimicrobial, antioxidant, and α-glucosidase inhibitory activity of the essential oil of *C. ferrugineus*. The results show that it inhibits α-glucosidases by exhibiting an IC_50_ value of 146 µg/mL. Among the major compounds present in the essential oil are caryophyllene, myrcene, β-phellandrene, germacrene D, linalool, and α-humulene [26].

### 2.5. C. gratissimus Burch. var. gratissimus

Common name: “koriba”. Species native to Africa. It is grown in towns and villages in Nigeria and the leaves are used to prepare water decoctions that are used as vasorelaxants and antihypertensives [52]. According to Okokon et al., the leaf decoction is also used to treat diabetes; therefore, the hypoglycemic effect of the ethanolic extract of the leaves was assessed in alloxan-induced hyperglycemic rats. The results show that the extract exerts a significant acute hypoglycemic effect from the first hour after its administration (200 mg/kg) and a sustained significant effect after seven days of treatment at the same dose [27]. Subsequent studies demonstrate that this species also diminishes blood glucose levels and improves the lipid profile in STZ-induced hyperglycemic rats [28,29]; however, it has been reported that the methanol and aqueous extract of the seed promote a significant increase in hepatic enzyme activities, and the accumulation of fatty acids and necrosis in the liver, which could indicate a potential toxic effect [53]. Finally, the aqueous extract of the leaves and its *n*-butanol fraction demonstrate a cardioprotective effect on toxicity generated by carbon tetrachloride in rat hearts [30].

### 2.6. C. grewioides Baill

Common name: “canela-de-cunhã”. Species native to Bolivia, eastern Brazil, and Peru. The entire plant is popularly used in the Brazilian TM to prepare a decoction [54] that is mainly drunk to treat nervous system disorders and induce appetite (anti-anorexigenic) [10,52]. Taking advantage of its properties on the central nervous system, Silva-Alves et al. tested the effect of the essential oil of *C. grewioides* on autonomic diabetic neuropathy, a complication developed by people with diabetes, in STZ-induced hyperglycemic rats, specifically on the DM-induced vagus nerve abnormalities. The results show that essential oil administration for four weeks (300 mg/kg/day; p.o.) prevents the decrease in the excitability and conduction velocity of the myelinated fibers of the vagus nerve without improving the hyperglycemic state of the rats [31]. Therefore, this species could be used for the complementary treatment of diabetes along with a hypoglycemic agent to correct both the diabetic complications and the hyperglycemia.

### 2.7. C. guatemalensis Lotsy

Common name: “copalchí”. Species native to Mexico and Central America. It is used in Guatemala to treat diabetes by the Cakchiquels. The bark is prepared as an infusion and four cups are drunk through the day [55]. Andrade-Cetto et al. assessed the acute hypoglycemic effect of the aqueous and ethanol–water extracts of the *C. guatemalensis* bark in STZ-NA-induced hyperglycemic rats. The latter had better outcomes by decreasing blood glucose levels by 42% after 90 min at a dose of 30 mg/kg. In the same study, the authors evaluated the potential inhibitory effect of *C. guatemalensis* on postprandial hyperglycemia and the activity of α-glucosidases. However, the ethanol–water extract was unable to reduce the hyperglycemic peak after a maltose load and had no inhibitory effect on rat intestinal α-glucosidase enzymes. Therefore, it was concluded that the hypoglycemic effect of this species was not related to this mechanism of action [32].

Regarding the phytochemical composition of ethanol–water extract, the presence of five *ent*-clerodane diterpenoids (junceic acid, crotoguatenoic acid A, crotoguatenoic acid B, formosin F, and bartsiifolic acid), and three flavonoids (rutin, epicatechin, and quercetin) was reported [33].

### 2.8. C. heterodoxus Baill

Common name: no records available. Species native to Brazil with no ethnopharmacological report on any medicinal use; however, its potential effects on glucose metabolism were studied because of its chemical richness in terpenoids, which have been shown to modulate mechanisms involved in insulin secretion and sensitivity [34]. Three terpenoids were isolated and characterized from the hexane fraction of the ethanol–water extract of the aerial parts and tested in any of the previously mentioned mechanisms.

Compound **1**: 3β-hydroxyhop-22(29)ene (Figure 1c). This compound decreased the postprandial peak by 35% at a dose of 1 mg/kg after 15 min in oral glucose tolerance tests (OGTTs), as well as increased insulin and glucagon-like peptide-1 (GLP-1) levels in rats co-administrated with 10 mg/kg of the compound and an oral glucose load. Moreover, this triterpene significantly increased the glycogen content of liver rats, stimulated glucose uptake, and promoted insulin vesicle translocation to the plasma membrane through the involvement of potassium and calcium channels and the modulation of calcium influx via protein kinase A (PKA) and protein kinase C (PKC) activation in pancreatic β-cells [34];Compound **2**: fern-9(11)-ene-2α,3β-diol. This triterpene was shown to have an acute antihyperglycemic effect in OGTTs at a dose of 10 mg/kg, diminishing the hyperglycemic peak by 15% in rats. In this same study, the authors synthetized a p-nitrobenzoyl derivative of this compound (Figure 1d) that exhibited a better antihyperglycemic effect at a dose of 1 mg/kg, inhibiting the hyperglycemic peak by 36% after the glucose load. This decrease was associated with an increase in insulin secretion since the derivative increased insulin concentration by 30% compared with the control group 15 min after the glucose load. In addition, the effect of the derivative compound on this mechanism was investigated in vitro in pancreatic islets, displaying an increased glucose uptake by 310% and influx of calcium by 360% compared with the controls. All these results suggest that this derivative compound has a direct effect on pancreatic β-cells, by promoting insulin secretion. One of the pathways involved in the promotion of insulin secretion is PKC activation; therefore, an inhibitor of this kinase (stearoylcarnitine chloride) was used to verify that the effects observed by the compound were due to the activation of PKC. The results show that the inhibitor abolishes the stimulatory effect of the compound on calcium influx and, as a result, it is concluded that the derivative triterpene stimulates insulin secretion through PKC activation [35];Compound **3**: 2α,3β,23-trihydroxyolean-12-ene (Figure 1e). The compound was shown to have the best antihyperglycemic effect at a dose of 10 mg/kg in OGTTs in rats where insulin levels were also quantified. The compound inhibited the hyperglycemic peak by 30% after the first 15 min of the study and enhanced the glucose-stimulated insulin secretion (GSIS) at 30 min after the glucose load. Additionally, it promoted a four-fold increase in glycogen content measured in rat livers and glucose uptake in isolated adipocytes via GLUT4 translocation [36].

### 2.9. C. hirtus L’Hér

Common name: no records available. This species has a wide distribution in Central and South American countries and has been introduced into African and Asian countries [56]. Some compounds isolated from this species are bis-nordolabradane, dolabradanes, kauranes, cyclopropakauranes, hirtusanes, and germacradiene esters [37].

Although there are no ethnopharmacological works that support its medicinal effects, the potential anti-inflammatory effect of the methanolic extract of the whole plant was analyzed since metabolites isolated from other *Croton* species demonstrated antihypertensive, anticancer, anti-inflammatory, antimicrobial, and antiviral properties. Kim et al. showed suppressed lipopolysaccharide-induced production of nitric oxide and inflammatory cytokines, such as interleukin 6 and tumor necrosis factor α, in RAW264.7 macrophages [38]. These results suggest that this species could prevent the underlying inflammation that promotes the development of illnesses such as diabetes, cardiovascular disease, and cancer.

### 2.10. C. klotzschianus (Wight) Thwaites

Common name: no records available. Species native to India and Sri Lanka. The traditional use of this species to treat diabetes in the Ayurvedic medicine has been reported [57]; however, it is a relatively uninvestigated medicinal plant. Govindarajan et al. showed the antihyperglycemic effect of the ethanol–water extract of the aerial parts at a dose of 300 mg/kg in normoglycemic rats by avoiding the postprandial peak observed at 30 min in OGTTs. Additionally, the hypoglycemic effect of this extract was demonstrated by decreasing glucose levels by 44.3% after 21 days of daily administration in STZ-induced hyperglycemic rats. The authors suggested that these effects could be explained by the stimulation of insulin secretion. The ethanol–water extract was shown to promote a 13-fold increase in insulin secretion at low glucose concentration (2 nM) in the MIN6 cell line. Likewise, the extract potentiated GSIS (20 nM) by promoting an eight-fold increase in insulin levels [39]. In the same work, the presence of quercetin and quinic acid in the aerial parts was reported.

### 2.11. C. krabas Gagnep

Common name: “Fai Nam”. Species native to Cambodia, Laos, Thailand, and Vietnam. There is no record on its traditional use; however, Rajachan et al. tested its potential bioactivity against diabetes because crude extracts of other species of *Croton* have been proven to have similar outcomes. In this study, the compound 12-oxohardwickiic acid and the new *ent*-clerodane diterpenoid crotonkrabas A (Figure 1f), isolated from the stems, showed an inhibitory activity on α-glucosidase enzymes (IC_50_ values of 155 and 404 μM, respectively), suggesting a potential antihyperglycemic effect [40].

### 2.12. C. lechleri Müll.Arg

Common name: “Sangre de Drago”. Species native to South America, whose bark releases a blood-red sap that is used to stop bleeding, accelerate wound healing, and as an anti-inflammatory agent [10,52]. Due to its cultural association with blood-related therapeutic properties, the latex was assessed on atherosclerotic and cardiovascular risk factors, which are involved in the progression of diabetes complications [41]. In this work, Chen et al. found that *C. lechleri* sap inhibited glucose-induced glycation of bovine serum albumin in a concentration-dependent manner (5 to 50 mg/mL). On the other hand, they observed a concentration-dependent decrease in human low-density lipoprotein cholesterol (LDL) oxidation and inhibition of the DPPH radical with an IC_50_ of 2.74 µg/mL, which could be due to the high content of polyphenols in the sap. In addition, the sap was shown to counteract H_2_O_2_-induced ROS formation in human umbilical vein endothelial cells. Taken together, these results demonstrate the potential use of *C. lechleri* sap in ROS-related diseases such as atherosclerosis.

### 2.13. C. membranaceus Müll.Arg

Common name: “Côte d’Ivoire”. Species native to west tropical Africa. The roots and the leaves are used as anti-inflammatory remedies to treat urinary retention caused by an enlarged prostate (benign prostatic hyperplasia; BPH) [12,52]. Some compounds reported from the roots of this species are: scopoletin, crotomembranafuran, julocrotine, N[N-(2-methylbutanoyl) glutaminoyl]-2-phenylethylamine, DL-threitol, gomojoside H, β-sitosterol, and β-sitosterol-3-D-glucoside [12].

Since an association between BPH and diabetes due to a common pathogenic mechanism (OxS) is suggested, the antihyperglycemic effect of root extract was assessed [42]. Briefly, the ethanolic extract of the roots demonstrated hypoglycemic properties by decreasing acute and chronic hyperglycemia by 60% in STZ-induced hyperglycemic rats at the maximum dose of 600 mg/kg. On the other hand, the aqueous extract of the roots also improved the cardiovascular profile by significantly diminishing low-density lipoprotein cholesterol and C-reactive protein and increasing high-density lipoprotein cholesterol in spontaneously hypertensive rats at a dose of 100 mg/kg/day for 60 days. In addition, blood glucose levels of *db*/*db* mice were significantly reduced after 3 h of the administration of the extract at a dose of 250 mg/kg [43].

### 2.14. C. persimilis Müll.Arg

Common name: “chucka”. Species native to Asian countries such as India, China, and Thailand. The root bark is used to treat liver ailments (chronic hepatitis and enlarged liver) [10,52]. According to Srisongkram et al., one of the main approaches to decrease postprandial hyperglycemia is reducing the α-amylase and α-glucosidase activity and traditional Thai medicine has several medicinal plants that have been shown to inhibit these enzymes; therefore, twenty-nine plants were selected to screen for potential inhibitors of α-glucosidase and α-amylase enzymes. The results show that the ethanolic extract of *C. perisimilis* stem exhibits a high inhibitory activity against the α-glucosidase enzyme, showing the lowest IC_50_ value (0.49 μg/mL) [44].

### 2.15. C. thurifer Kunth

Common name: “mosquera”. Species native to the Amazonian region (Brazil, Ecuador, and Peru). The latex is used to treat wounds, sores, and ulcers. Considering the phytochemical richness of *Croton* species and their usefulness as medicinal plants, Morocho et al. tested *C. thurifer* as a potential source of hypoglycemic agents. Among all the isolated compounds obtained from the organic extracts of the leaves, the ethyl acetate extract, as well as trans-tiliroside and (3R, 20S)-3-acetoxy-20-hydroxydammar-24-ene (Figure 1g), displayed a good inhibitory activity on α-glucosidases with IC_50_ values of 1.77 mg/mL, 114.85 µM, and 292.87 µM, respectively [45].

### 2.16. C. tiglium L.

Common name: “croton-oil plant”. Species native to the tropical and sub-tropical Asian countries. This species has several medicinal uses in Ayurveda, Unani, Siddha, and Chinese medicine as a laxative, purgative, and to treat tumors and cancerous sores [10,52]. In addition, the seed oil is used for hypertension and inflammation [58]. Since it has been documented that medicinal plants of Myanmar have properties against diabetes, some species of this region were selected to investigate their effect on porcine pancreatic lipase inhibition and glucose uptake in 3T3-L1 adipocyte assays. The ethanolic extract of *C. tiglium* seeds showed the highest increase in glucose uptake at a concentration of 40 μg/mL, which could indicate a potential insulin-sensitizing effect [46].

### 2.17. C. yunnanensis W.W.Sm

Common name: “ji gu xiang”. Species native to China whose roots are used to treat cholera, dysentery, and an inflamed and painful throat [52]. Even though there is no ethnopharmacological report on diabetes, Jiang et al. tested the potential insulin-sensitizing activity of some isolated compounds from the roots due to the structural and bioactive diversity of *Croton* diterpenes on glucose metabolism, as it has been reported for other species of the genus [48]. In this work, nine new diterpenes were isolated from the ethanolic extract of the roots: eight neo-clerodane diterpenes and one 15,16-dinor-ent-pimarane diterpene. Most of these compounds promoted glucose uptake activity in insulin-resistance 3T3-L1 adipocytes; however, only crotonine A (Figure 1h) and crotonine F (Figure 1i) demonstrated an ability to potentiate the glucose uptake effect in the presence of low insulin, suggesting an important role in the insulin signaling pathway.

## 3. Discussion

TM plays an important role in the treatment of diseases worldwide. Historically, it has been used to maintain health and to prevent and treat illnesses, particularly chronic diseases [8]. Despite the scientific progress in conventional medicine, TM practice is also growing because of the increasing prevalence of chronic conditions, rising costs of essential health services, and low adherence to allopathic treatments. As a result, nonconventional medicine is being increasingly used to treat chronic noncommunicable diseases, such as T2D [7,59]. In addition to TM, which is used as a lead in the ethnopharmacological approach in drug discovery, the search for bioactive compounds by chemical similarity based on taxonomy- and chemotaxonomy-guided random screening is also making a major contribution in the development of new drugs [60]. The results of our review of the literature show that both approaches are being applied in the search for drugs for T2D from, or based on, *Croton* species.

In general, it was found that there are many *Croton* plants used in TM that have limited scientific studies supporting their usage as they only address their potential hypoglycemic or antihyperglycemic effects, while for other species that have works on specific mechanisms, the information on their traditional usage is incomplete or not related to diabetes. Thus, it is suggested to perform ethnopharmacological studies on these species to obtain accurate traditional information, such as common names, current herbal uses, ways of preparation, parts and dosages used, to enrich the knowledge that is already available in order to carry out more precise pharmacological studies on their hypoglycemic mechanisms.

According to the results obtained, there are three species whose traditional usage is not reported in the literature (*C. heterodoxus*, *C. hirtus*, and *C. krabas*); nevertheless, they demonstrated good effects in reducing postprandial hyperglycemia and inflammation, suggesting that they can be used for the management of T2D. In this sense, it is recommended to carry out ethnobotanical studies. On the other hand, the pharmacological targets tested for 11 of the species found were not related to traditional use, namely, other kinds of extractions and different plant parts were used than those reported in previous ethnobotanical studies. Moreover, although they had field studies, several of them were not reported to be used for treating diabetes. In those specific cases, bioprospecting or a chemotaxonomic approach was used to justify the search for mechanisms of action on diabetes.

Of all the reported plants, only three species had a positive correspondence between their traditional use and the pharmacological tests carried out, i.e., the type of extract, the plant part, the doses, and the tested target. These species were *C. cajucara*, *C. guatemalensis*, and *C. lechleri*. It is important to note that there are only studies on their hypoglycemic and antihyperglycemic effects, so it is necessary to delve into their specific modes of action, especially for *C. cajucara* and *C. guatemalensis*.

Another aspect to discuss is the doses used in the pharmacological tests, which directly impact the outcomes. When the full extracts were tested, generally high doses were used that possibly did not correlate with traditional doses, as the ethnobotanical information was unclear. However, *C. cajucara* and *C. guatemalensis* stood out in this regard, since the doses used in their pharmacological studies corresponded to the information obtained from ethnobotanical reports. Specifically, *C. cajucara* was the most referenced species used in TM for hyperglycemia. The extract of this plant, as well as its main compound t-DCTN (Figure 1b), showed a marked hypoglycemic effect, although the characterization of their mechanisms of action is still unclear.

The multifactorial etiopathogenesis of T2D makes it a complex disease, and its treatment has typically required intervention with several single-target drugs. However, the search for multitarget drugs is increasingly relevant because disease management through various pathways could promote better glycemic control to achieve the goals recommended by health institutions [61]. In this context, natural products and medicinal plants have been shown to have multiple mechanisms of action that ameliorate three or more of the pathophysiological features contributing to the disease [62,63].

One of the underlying factors that promote the development of T2D in its early stages is obesity-induced inflammation. These inflammatory processes cause metabolic abnormalities such as dyslipidemia, hyperinsulinemia, and IR through mechanisms such as endothelial dysfunction, free radical production, lipid peroxidation, and proinflammatory cytokine production [64]. These metabolic disturbances ultimately decrease the insulin response in insulin-sensitive tissues and insulin synthesis in pancreatic β-cells [65]. As the disease progresses, chronic hyperglycemia resulting from IR generates mitochondrial dysfunction and ROS through PKC activation. Increased ROS impair the function of antioxidant system enzymes, such as superoxide dismutase, glutathione peroxidase, and catalase, further contributing to OxS and the development of vascular inflammation, endothelial dysfunction, and atherosclerosis over the long term [6,66].

The South American species *C. grewioides*, *C. hirtus*, and *C. lechleri* have been shown to decrease diabetic complications, inflammation, and OxS by protecting the myelinated vagus nerve fibers, reducing nitric oxide and cytokine production, and decreasing bovine serum albumin glycation, LDL oxidation, and H_2_O_2_-induced OxS, respectively. Therefore, they can be used as complementary treatments for the underlying inflammation and OxS, as well as complications of the disease. However, in addition to having antioxidant and anti-inflammatory properties, other *Croton* species, such as *C. bonplandianus*, *C. cajucara*, *C. ferrugineus*, *C. gratissimus* var. *gratissimus*, and *C. membranaceus*, which could provide better management due to their combined effects, also exhibit hypoglycemic, antihyperglycemic, antilipidemic, and antihypertensive effects.

When the disease is established, hyperglycemia in patients with diabetes originates from two sources. One source is via an endogenous pathway, in which IR in the liver leads to uncontrolled glucose production and subsequent fasting hyperglycemia. The other source is via an exogenous pathway, in which several factors, namely, intestinal glucose absorption, impaired insulin secretion, incretin dysfunction, and peripheral IR, play essential roles in the generation of postprandial hyperglycemia [67]. However, none of the reported species were evaluated for any mechanism that affects fasting hyperglycemia, although several plants were shown to decrease fasting glucose levels in vivo either acutely or chronically, i.e., *C. cajucara* and its main compound t-DCTN, *C. cuneatus*, *C. gratissimus* var. *gratissimus*, *C. guatemalensis*, and *C. membranaceus*. Therefore, we recommend further studies on the potential inhibitory effect of these species and their main actions on mechanisms involved in the generation of fasting hyperglycemia, such as hepatic glucose production.

After a meal, multiple mechanisms are involved in the regulation of postprandial glucose levels, which are reportedly impaired in T2D patients. This dysregulation can lead to postprandial hyperglycemia (>140 mg/dL), which has a strong association with cardiovascular risk factors [68]. From an ethnopharmacological point of view, decoctions and infusions of medicinal plants used for T2D can also be drunk before each meal [69], which could avoid the postprandial hyperglycemic peaks that are generally too high in people with DM [70]. In this regard, species such as *C. bonplandianus*, *C. ferrugineus*, *C. heterodoxus*, *C. klotzschianus*, *C. krabas*, *C. persimilis*, *C. thurifer*, *C. tiglium*, and *C. yunnanensis* have the potential to manage postprandial hyperglycemia, as all of them have studies on some of the mechanisms involved in this metabolic state: inhibition of carbohydrate hydrolysis and promotion of glycogen synthesis, insulin secretion, and glucose uptake (Figure 2). However, not all of them have ethnobotanical studies that support their traditional use, namely, the works that have been published on their benefits on glucose metabolism have been carried out due to their chemical similarity with other *Croton* species.

The most frequently evaluated target mechanism has been the inhibition of α-glucosidase enzymes, which are located on the border of the intestinal brush and are responsible for hydrolyzing polysaccharides into monosaccharides to facilitate their absorption. In T2D, the activity of these enzymes is one of the mechanisms responsible for the postprandial increase in glucose; therefore, inhibiting their activity may be an effective approach to decrease postprandial hyperglycemic peaks by delaying carbohydrate absorption after food intake [71]. The species identified with this inhibitory effect were *C. bonplandianus* and three of its isolated compounds (2 *H*-1-benzopyran-2-one, betulin (Figure 1a), and 3,5-dimethoxy 4-hydroxy cinnamic acid), *C. ferrugineus*, *C. krabas* and its isolated compound 12-oxohardwickiic acid, *C. persimilis*, and *C. thurifer* and its isolated compounds trans-tiliroside and (3R, 20S)-3-acetoxy-20-hydroxydammar-24-ene (Figure 1g). However, only in vitro studies were performed to evaluate these species and their compounds; thus, it is important to conduct in vivo studies to confirm their antihyperglycemic effect and their potential therapeutic benefits on postprandial hyperglycemia.

The second mechanism evaluated was the promotion of insulin secretion. The ethanol–water extract of *C. klotzschianus* and compounds isolated from *C. heterodoxus* have been shown to have an insulin secretagogue effect in vitro and an antihyperglycemic effect in vivo in OGTTs. This mechanism can be enhanced directly by promoting insulin vesicle translocation or indirectly by increasing incretin levels. The incretin effect potentiates GSIS and influences whole-body glucose metabolism through the action of hormones secreted by endocrine intestinal cells: GLP-1 and glucose-dependent insulinotropic polypeptide (GIP). These two incretins stimulate insulin secretion by generating a positive feedback system that allows a rapid and appropriate response by pancreatic β-cells, controlling plasma glucose levels after food consumption [72]. Some authors point out that the incretin effect is diminished in people with T2D because pancreatic β-cells do not respond adequately to these hormones. Hence, the application of incretin therapy results in a sustained improvement not only in glycemic levels but also in body weight. In addition, incretin treatment has a positive impact on inflammation, cardiovascular and liver health, and the central nervous system, especially improving sleep apnea [73,74,75]. According to the literature search, only the triterpene 3β-hydroxyhop-22(29)ene (Figure 1c), isolated from *C. heterodoxus*, increased insulin secretion through GLP-1, which was associated with the antihyperglycemic effect exhibited in the OGTTs in vivo.

Other mechanisms involved in the generation of postprandial hyperglycemia that were evaluated were the promotion of glycogen synthesis and peripheral glucose uptake via GLUT4 translocation. Both processes are crucial to regulate blood glucose levels in the postprandial state [76]; thus, the amelioration of their function may lead to better control of hyperglycemia in T2D patients. The triterpene 2α,3β,23-trihydroxyolean-12-ene (Figure 1e), isolated from *C. heterodoxus*, increased both mechanisms in addition to promoting insulin secretion. Moreover, the ethanolic extract of *C. tiglium* seeds and terpenoids crotonine A and crotonine F (Figure 1h,i) from *C. yunnanensis* increased glucose uptake in 3T3-L1 adipocytes, showing a potential insulin-sensitizing effect that needs to be confirmed in subsequent in vivo studies.

Despite the diversity of proven mechanisms of action of the genus *Croton*, some of the species reported, such as *C. cajucara*, *C. cuneatus*, *C. gratissimus* var. *gratissimus*, *C. guatemalensis*, and *C. membranaceus*, have only been studied for their hypoglycemic and antihyperglycemic effects in vivo; thus, more work is needed to elucidate the mechanisms of action by which these species exert these effects.

Regarding their chemical composition, most of the listed species only have basic phytochemical studies without further delving into the mechanisms of action of the main components. Among the phytochemicals identified, triterpenoids and diterpenoids are the predominant metabolites that have been reported (Figure 1), and their scaffolds have been attributed to a wide variety of biological activities. For instance, the triterpenoids betulin and (3R, 20S)-3-acetoxy-20-hydroxydammar-24-ene (Figure 1a,g) show inhibitory activity against α-glucosidase enzymes. In this context, it has recently been reported that cores such as steroid skeletons exert a major effect on these oligosaccharide-degrading enzymes, making them good starting points for the synthesis of inhibitors [77]. Likewise, the pentacyclic triterpenes isolated from *C. heterodoxus* were shown to regulate postprandial hyperglycemia through various stimulatory mechanisms of insulin secretion, which is in line with other similar compounds, such as oleanolic acid and 20(*S*)-ginsenoside Rg_3_ [78,79]. Therefore, these types of phytochemicals can modulate ionic channels, activate protein kinase signaling, and stimulate secretory machinery in pancreatic β-cells.

Furthermore, some pimaranes and clerodanes, two classes of diterpenes isolated from *C. yunannensis*, promoted glucose uptake in vitro in 3T3-L1 adipocytes, especially crotonines A and F (Figure 1h,i), which showed a potentiated effect when were tested with insulin. The molecular mechanism of this type of diterpenoid is the activation of the insulin receptor–Akt pathway in vivo [11]; thus, these scaffolds may be related to the decrease in IR.

Taken together, these findings emphasize the importance of the *Croton* genus both in TM and in the search for starting points for new molecules with effective mechanisms of action for the treatment of T2D. However, the progress that has been made to date is in its initial stages; only the hypoglycemic effects of the full extracts and a few compounds have been described. We recommend further research that continues to search for new mechanisms of action that explain how both full extracts and their main phytochemicals exert their therapeutic effects on diabetes.

Based on our experience from previous studies performed both in the field and in the laboratory, we reached the consensus that most of the plants used to treat DM are taken as “agua de uso” [55], which consists of making an infusion with approximately 20 g of plant (a fist) that is drunk throughout the day between meals. The effectiveness of this mode of consumption relies on the fact that hepatic glucose production, specifically gluconeogenesis, is the pathway that contributes the most to fasting hyperglycemia in patients with DM [80]; consequently, it could improve fasting glucose levels by inhibiting the glucose output by the liver [81]. None of the plants reported in this review have studies on the inhibition of this mechanism; therefore, we strongly encourage addressing this mode of action in future work.

## 4. Methods

A search of the literature was performed based on the methodology of the preferred reporting items for systematic review and meta-analysis (PRISMA) [82] in PubMed, Scopus, and Web of Science, all highly recognized databases, with the keywords “Croton”, “diabetes”, “hypoglycemic”, and “antihyperglycemic”. All articles generated by the bibliographic search that met the inclusion criteria, which covered the years 1981 to 2023, were considered (Figure 3). The fully accepted scientific names and distribution of each species were obtained from the webpages of the Tropicos database [83], the World Flora Online [56], and Plants of the World Online by the Royal Botanic Gardens, Kew [84].

## 5. Conclusions

This research report highlights the importance of *Croton* species for the treatment of T2D by reviewing the literature on their mechanisms of action (Figure 2). All these species may contribute to adequate management of the disease and, thus, delay the onset of its characteristic complications, which are responsible for the comorbidities affecting the quality of life of people with diabetes. We recommend further studies on *C. cajucara*, *C. cuneatus*, *C. gratissimus* var. *gratissimus*, *C. guatemalensis*, and *C. membranaceus* to elucidate the mechanisms responsible for their hypoglycemic and antihyperglycemic effects, particularly on those that generated fasting hyperglycemia, since all these species were shown to decrease blood glucose levels in this metabolic state in previous studies.

## Figures and Tables

**Figure 1 plants-12-02014-f001:**
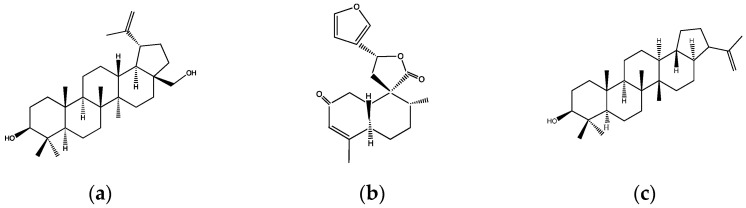
Chemical structures of terpenoids identified in *Croton* species with reported diabetes-related pharmacological activities: (**a**) betulin (*C. bonblandianus*); (**b**) trans-dehydrocrotonin (*C. cajucara*); (**c**) 3β-hydroxyhop-22(29)ene, (**d**) p-nitrobenzoyl derivative of fern-9(11)-ene-2α,3β-diol, and (**e**) 2α,3β,23-trihydroxylean-12-ene (*C. heterodoxus*); (**f**) crotonkrabas A (*C. krabas*); (**g**) (3R, 20S)-3-acetoxy-26-hydroxydammar-24-ene (*C. thurifer*); (**h**) crotonine A and (**i**) crotonine F (*C. yunnanensis*).

**Figure 2 plants-12-02014-f002:**
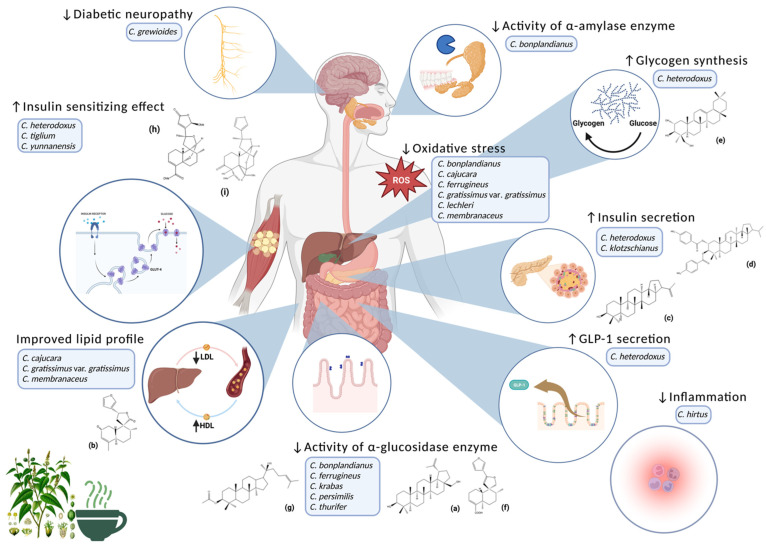
Summary of the mechanisms of action involved in the diabetes-related pharmacological effects of *Croton* species. The modes of action involved in the modulation of glucose and lipid metabolism, as well as inflammation, oxidative stress and diabetic complications of *Croton* species and their compounds are schematized. *Croton* image obtained from Wikipedia. Created with Biorender.com.

**Figure 3 plants-12-02014-f003:**
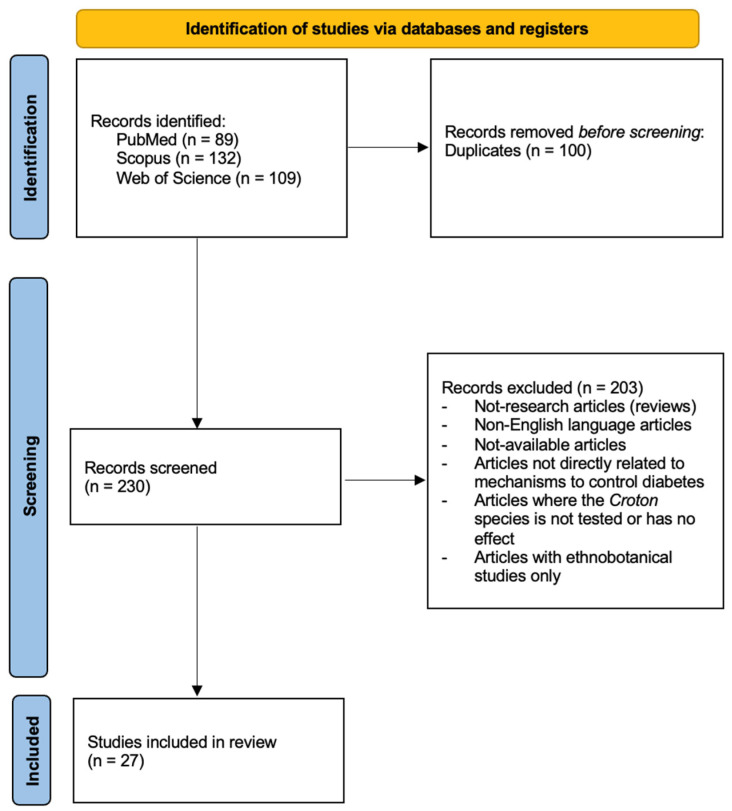
PRISMA flowchart.

**Table 1 plants-12-02014-t001:** *Croton* species with pharmacological reports on diabetes or related diseases.

Species	Common Names	Places Where It Is Used	Type of Extracts and Parts Used	Identified Compounds	Pharmacological Studies Related to Diabetes	References
*C. bonplandianus* Baill.	“Escoba negra” “Ban tulsi”	South America (Argentina) Asia (India)	Dichlorometane extract of the whole plant. Chloroform fraction obtained from methanol extract of the leaves.	2 *H*-1-benzopyran-2-one Betulin (Figure 1a) 3,5-Dimethoxy 4-hydroxy cinnamic acid Z-5-Nonadecene Cyclotetracosane N-Nonadecenol-1 Cycloeicosane 3-eicosene Z-8—Hexadecene 6-5heptadecenal Phenol,2,4-bis(1,1-dimethyl)	In vitro: Inhibition of α-glucosidase and α-amylase activities. In vitro: Antioxidant activity.	[16,17,18,19]
*C. cajucara* Benth.	“Sacaca”	South America (Brazil)	Aqueous extract of the bark.	Trans-Crotonin trans-Dehydrocrotonin (Figure 1b) Acetyl aleuritolic acid	In vivo: Hypoglycemic and antihyperglycemic effects of t-DCTN in alloxan-induced hyperglycemic rats. In vivo: Antihypertriglyceridemic effect of t-DCTN in ethanol-induced hypertriglyceridemic rats. In vivo: Antihypertensive effect of t-DCTN in normotensive rats. In vivo: Reduction in hepatic OxS markers in STZ-induced hyperglycemic rats.	[10,20,21,22,23,24]
*C. cuneatus* Klotzsch	“Arapurima”	South America (Venezuela)	Aqueous extract of the stem bark.	A-11 Eudesmene Methyleugenol 4-α-Seleniol Cedryl propyl ether τ-Cadinol Cubenol	In vivo: Hypoglycemic effect in STZ-induced hyperglycemic rats.	[25]
*C. ferrugineus* Kunth	“Mosquera”	South America (Ecuador)	Essential oil of the leaves.	Caryophyllene Myrcene β-Phellandrene Germacrene D Linalool α-Humulene	In vitro: Inhibition of α-glucosidase activity. In vitro: Antioxidant activity.	[26]
*C. gratissimus* Burch. Var. *gratissimus*	“Koriba”	Africa (Nigeria)	Ethanolic and aqueous extract of the leaves. Aqueous extract of the leaves.	Trachylobane *ent*-trachyloban-3β-ol *ent*-18-Hydroxy-trachyloban-3-one Isopimara-7,15-dien-3β-ol *ent*-18-hydroxytrachyloban-3β-ol *ent*-18-hydroxyisopimara-7,15-diene-3β-ol	In vivo: Acute and chronic hypoglycemic effects in alloxan-induced hyperglycemic rats. In vivo: Improvement in lipid profile in STZ-induced hyperglycemic rats. In vivo: Cardioprotective effect on carbon tetrachloride-induced cardiac toxicity in rats. In vivo: Decreased lipid peroxidation and antioxidant markers in STZ-induced hyperglycemic rats.	[10,27,28,29,30]
*C. grewioides* Baill.	“Canela-de-cunhã”	South America (Brazil)	Essential oil of the leaves.	Anethole Estragole	In vivo: Protection against decreased excitability and conduction velocity of myelinated vagus nerve fibers in STZ-induced hyperglycemic rats.	[31]
*C. guatemalensis* Lotsy	“Copalchí”	Central America (Guatemala)	Aqueous and ethanol-water extracts of the bark.	Junceic acid Crotoguatenoic acid A and B Bartsiifolic acid Formosin F Rutin Epicatechin Quercetin	In vivo: Acute hypoglycemic effect in STZ-NA-induced hyperglycemic rats.	[32,33]
*C. heterodoxus* Baill.	No records available	No reports on medicinal uses	Ethanol–water extract of the aerial parts.	3β-Hydroxyhop-22(29)ene (Figure 1c) p-nitrobenzoyl derivative of fern-9(11)-ene-2α,3β-diol (Figure 1d) 2α,3β,23-Trihydroxyolean-12-ene (Figure 1e)	In vivo: Decreased postprandial hyperglycemia due to increased insulin and/or GLP-1 levels in OGTTs performed in healthy rats. In vivo: Increased glycogen content in liver rats. In vitro: Promoted glucose uptake in pancreatic islets. In vitro: Promoted insulin vesicle translocation via PKA and/or PKC activation in pancreatic islets. In vitro: Promoted glucose uptake via GLUT4 translocation in adipocytes.	[34,35,36]
*C. hirtus* L’Hér.	No records available	No reports on medicinal uses	Methanol extract of the whole plant.	Bis-nordolabradane Dolabradanes Kauranes Cyclopropakauranes Hirtusanes Germacradiene esters	In vitro: Suppressed lipopolysaccharide-induced production of nitric oxide and inflammatory cytokines in RAW264.7 macrophages.	[37,38]
*C. klotzschianus* (Wight) Thwaites	No records available	Asia (India)	Ethanol–water extract of the aerial parts.	Quercetin Quinic acid	In vivo: Decreased postprandial hyperglycemia in OGTTs performed in healthy rats. In vivo: Sub-chronic hypoglycemic effect in STZ-induced hyperglycemic rats. In vitro: Promoted insulin secretion at low glucose concentration and enhanced GSIS in MIN6 cell line.	[39]
*C. krabas* Gagnep.	“Fai Nam”	Asia (Thailand)	*n*-Hexane and ethyl acetate extracts of the stems.	Crotonkrabases A–C (crotonkrabas A: Figure 1f) 12-Oxohardwickiic acid Crotonpyrone B	In vitro: Inhibition of α-glucosidase activity.	[40]
*C. lechleri* Müll.Arg.	“Sangre de Drago”	South America (Ecuador)	Bark sap	Crolechinol Crolechinic acid Taspine 3′,4-*O*-dimethylcedrusin Proanthocyanidin SP–303	In vitro: Decreased bovine serum albumin glycation, LDL oxidation, and H_2_O_2_-induced OxS in human umbilical vein endothelial cells.	[10,41]
*C. membranaceus* Müll.Arg.	“Côte d’Ivoire”	Africa (Ghana)	Ethanolic and aqueous extracts of the roots.	Scopoletin Crotomembranafuran Julocrotine N[N-(2-methylbutanoyl) glutaminoyl]-2-phenylethylamine DL-threitol Gomojoside H β-Sitosterol β-Sitosterol-3-D-glucoside	In vivo: Acute and chronic hypoglycemic effects in STZ-induced hyperglycemic rats. In vivo: Improved cardiovascular profile in spontaneously hypertensive rats. In vivo: Acute hypoglycemic effect in db/db mice. In vitro: Antioxidant activity.	[12,42,43]
*C. persimilis* Müll.Arg.	“Chucka”	Asia (Thailand)	Ethanol–water extract of the stem.	11-Dehydro (-) hardwickiic acid Labda-7,12(*E*),14-triene Labda-7,12(*E*),14-triene-17-al Labda- 7,12(*E*),14-triene-17-ol Labda-7,12(*E*),14-triene-17-oic acid Crotocembranoic acid Neocrotocembranal	In vitro: Inhibition of α-glucosidase activity.	[10,44]
*C. thurifer* Kunth	“Mosquera”	South America (Ecuador)	Ethyl acetate extract of the leaves.	(3R, 20S)-3-Palmitate-20-hydroxydammar-24-ene (3R, 20S)-3-Acetoxy-20-hydroxydammar-24-ene (Figure 1g) *trans*-Phytol Vomifoliol β-Sitosterol trans-Tiliroside Sparsifol	In vitro: Inhibition of α-glucosidase activity.	[45]
*C. tiglium* L.	“Croton-oil-plant” “Kanakho”	Asia (India, China)	Ethanolic extract of the seeds.	12-*O*-isobutyrylphorbol-13-decanoate 12-*O*-(2-methyl)butyrylphorbol-13-octanoate 12-*O*-(2-methyl)butyrylphorbol-13-tiglate Crotignoid A-K 12-*O*-tiglylphorbol-4-deoxy-4β-phorbol-13-acetate 12-*O*-tiglylphorbol-4-deoxy-4β-phorbol-13-hexadecanoate	In vitro: Promoted glucose uptake in 3T3-L1 adipocytes.	[46,47]
*C. yunnanensis* W.W.Sm.	“Ji gu xiang”	Asia (China)	Ethanolic extract of the roots.	Crotonine A-I (crotonine A: Figure 1h, crotonine F: Figure 1i) Crotonyunnan A Hardwickic acid methyl ester 12-Hydroxyhardwickic acid methyl ester 12-Oxohardwickic acid methyl ester Sacacarin Phlorizin	In vitro: Promoted glucose uptake in insulin-resistant 3T3-L1 adipocytes.	[48]

t-DCTN: trans-dehydrocrotonin; OxS: oxidative stress; STZ: streptozotocin; NA: nicotinamide; GLP-1: glucagon-like peptide-1; OGTTs: oral glucose tolerance tests; PKA: protein kinase A; PKC: protein kinase C; GLUT4: glucotransporter 4; GSIS: glucose-stimulated insulin secretion; LDL: low-density lipoprotein cholesterol.

## Data Availability

Data sharing not applicable.

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
