# Peer review of "Diabetes-Related Mechanisms of Action Involved in the Therapeutic Effect of Croton Species: A Systematic Review"

_plants, 2023, doi:10.3390/plants12102014_

Round 1

Reviewer 1 Report

The review paper is focused on the chemical characterization and application of Croton genus for the treatment of chronic diseases. It has been emphasized that these terpenoid-rich species traditionally used to treat gastrointestinal diseases, heal wounds, and relieve pain have a wide range of therapeutic effects. Interestingly, up to 17 species with hypoglycemic, anti-hyperglycemic, anti-lipidemic, anti-hypertensive, antioxidant, and anti-inflammatory effects were found. On the other hand, no reports so far highlighted their action mechanisms for the treatment of diabetes.

The review is surely interesting, and it is expected that it contributes to the plant science knowledge of the genus, as well as to be used in future references on the identification and development of new molecules/phytomedicines that help in the treatment of diabetes.

Title. Avoid full stop!

1. Introduction, lines 78-82.  The authors are invited to enlarge this part by reporting the attempts (if any) that have been made for elucidating the action mechanisms on diabetes.

2. Methods. I invite the authors to insert the time frame considered for their literature search.

The review paper is focused on the chemical characterization and application of Croton genus for the treatment of chronic diseases. It has been emphasized that these terpenoid-rich species traditionally used to treat gastrointestinal diseases, heal wounds, and relieve pain have a wide range of therapeutic effects. Interestingly, up to 17 species with hypoglycemic, anti-hyperglycemic, anti-lipidemic, anti-hypertensive, antioxidant, and anti-inflammatory effects were found. On the other hand, no reports so far highlighted their action mechanisms for the treatment of diabetes.

The review is surely interesting, and it is expected that it contributes to the plant science knowledge of the genus, as well as to be used in future references on the identification and development of new molecules/phytomedicines that help in the treatment of diabetes.

Title. Avoid full stop!

1. Introduction, lines 78-82.  The authors are invited to enlarge this part by reporting the attempts (if any) that have been made for elucidating the action mechanisms on diabetes.

2. Methods. I invite the authors to insert the time frame considered for their literature search.

Author Response

Many thanks for your positive feedback.

  1. Avoid full stop!

We have corrected that error. Thanks for the observation.

  1. Introduction, lines 78-82.  The authors are invited to enlarge this part by reporting the attempts (if any) that have been made for elucidating the action mechanisms on diabetes.

We have included more details about this topic (lines 81-87).

  1. I invite the authors to insert the time frame considered for their literature search.

We have added the considered coverage time of the search for the review (lines 96-98).

Reviewer 2 Report

The manuscript should be revised for linguistic, grammatical, and style errors.

The title should include "Systematic review" 

The references should be updated 

At the 1st introductory sentence of the introduction, the authors used 2014-reference  (Newman et al. reported that 52 new chemical entities based 35 on plant sources with potential in the treatment of diabetes were identified between 1981 36 and 2014)

Newman, D.J.; Cragg, G.M. Natural Products as Sources of New Drugs from 1981 to 2014. J Nat Prod 2016, 79, 629–661, 548 doi:10.1021/acs.jnatprod.5b01055.

The authors should use recent statistics related to the use of medicinal plants and Diabetes Mellitus.

The potential mechanism of action of each plant species should be fully discussed in the text and within the summary table.

The manuscript should be revised for linguistic, grammatical, and style errors.

Author Response

Dear Reviewer Thanks for help us to improve the manuscript.

  1. The manuscript should be revised for linguistic, grammatical, and style errors.

We have reviewed and corrected the manuscript.

  1. The title should include "Systematic review" 

We have changed the title to include "systematic review".

  1. The references should be updated

Thank you for the suggestion. We have reviewed and updated the references throughout the manuscript.

  1. At the 1st introductory sentence of the introduction, the authors used 2014-reference  (Newman et al. reported that 52 new chemical entities based 35 on plant sources with potential in the treatment of diabetes were identified between 1981 36 and 2014): Newman, D.J.; Cragg, G.M. Natural Products as Sources of New Drugs from 1981 to 2014. J Nat Prod 2016, 79, 629–661, 548 doi:10.1021/acs.jnatprod.5b01055.

Thank you for the observation. We have updated this reference.

  1. The authors should use recent statistics related to the use of medicinal plants and Diabetes Mellitus.

We have added more recent references about this topic in the first paragraph of Introduction (lines 35-42).

  1. The potential mechanism of action of each plant species should be fully discussed in the text and within the summary table.

Thanks for the suggestion. We have enriched the discussion (lines 397-439) on this topic.

Reviewer 3 Report

The paper describes the use of extracts and/or compounds obtained from Croton species for diabetes treatment.  Although you have a good job to summarise the information, this is actually what this paper is, a summary... and not a review. There is no deeper critical assessment of data, and as submitted it is an encyclopedia presentation of the topic, albeit of high interest.  From my point of view, a good review should not only list facts but should also provide deeper interpretation and guide further work in the field.

Perhaps you can consider the following points:  

1. Has all the traditional uses for DM treatment been studied? 

2. What is the link between traditional uses and anti-diabetic properties reported? 

3. How do traditional use, phytochemistry, and pharmacology all tie in...does the data make sense? 

4. What is your interpretation of the assays and data...were the concentrations at which testing was done realistically?

The data shown in excellent Fig. 3 will be able to help you in this discussion.

The language-written quality is very good. However, check the text and make the minor change: hydroxy instead hidroxi.

Author Response

Dear Reviewer Thanks for your suggestions, certainly they helped us to improve the work. 

  1. The paper describes the use of extracts and/or compounds obtained from Crotonspecies for diabetes treatment.  Although you have a good job to summarise the information, this is actually what this paper is, a summary... and not a review. There is no deeper critical assessment of data, and as submitted it is an encyclopedia presentation of the topic, albeit of high interest.  From my point of view, a good review should not only list facts but should also provide deeper interpretation and guide further work in the field.

Perhaps you can consider the following points:  

  1. Has all the traditional uses for DM treatment been studied? 
  2. What is the link between traditional uses and anti-diabetic properties reported? 
  3. How do traditional use, phytochemistry, and pharmacologyall tie in...does the data make sense? 
  4. What is your interpretation of the assays and data...were the concentrations at which testing was done realistically?

The data shown in excellent Fig. 3 will be able to help you in this discussion.

Thank you for the suggestions. We have enriched the discussion taking these points into account (lines 397-426) and (565-574).

  1. The language-written quality is very good. However, check the text and make the minor change: hydroxy instead hidroxi.

We have reviewed and corrected these errors throughout the text.

Round 2

Reviewer 1 Report

The authors have adequately addressed all remarks and the paper can be now accepted in the present form.

Reviewer 2 Report

As the authors addressed the reviewers' comments, I suggest acceptance of the manuscript.

Reviewer 3 Report

All modifications suggested by the reviewers were implemented in the paper. Notably, the insertion of traditional uses and future perspectives enhanced the quality of the paper.  Therefore, I recommend the acceptance of the paper.